# Alteration of the Fecal but Not Salivary Microbiome in Patients with Behçet’s Disease According to Disease Activity Shift

**DOI:** 10.3390/microorganisms9071449

**Published:** 2021-07-06

**Authors:** Jin Cheol Kim, Mi Jin Park, Sun Park, Eun-So Lee

**Affiliations:** 1Department of Dermatology, Ajou University School of Medicine, Suwon 16499, Korea; kimjcmed@naver.com (J.C.K.); hijk999@naver.com (M.J.P.); 2Department of Microbiology and Immunology, Ajou University School of Medicine, Suwon 16499, Korea; sinsun@ajou.ac.kr

**Keywords:** Behçet’s disease, dysbiosis, microbiome, recurrent aphthous stomatitis, saliva, stool

## Abstract

The human microbiome plays an important role in various diseases, including Behçet’s disease (BD). However, the effects of disease activity and covariates influencing the microbial composition have not yet been investigated. Therefore, we investigated the fecal and salivary microbiomes of BD patients compared to those of recurrent aphthous ulcer (RAU) patients, as well as dietary habit-matched healthy controls (HCs) selected from immediate family members using 16S rRNA gene sequencing. The fecal microbiome alpha diversity of BD patients was not different from that of their matched HCs, although it was higher than that of unrelated HCs and decreased in BD patients with disease activity. A tendency toward clustering in the beta diversity of the fecal microbiome was observed between the active BD patients and their matched HCs. Active BD patients had a significantly higher abundance of fecal *Bacteroides uniformis* than their matched HCs and patients with the disease in an inactive state (*p* = 0.038). The abundance of salivary *Rothia mucilaginosa group* was higher in BD patients than in RAUs patients. BD patients with uveitis had different abundances of various taxa, compared to those without uveitis. Our results showed an association of fecal microbiome composition with BD disease activity and symptoms, suggesting the possible role of the gut microbiome in BD pathogenesis.

## 1. Introduction

The human microbiome plays an important role in health and disease, including inflammatory and autoimmune diseases. In healthy people, a symbiosis exists between the body and the microbiome, and this is maintained by complex and dynamic interactions [1,2]. Multiple factors, such as dietary habits, sex, age, genotype, and exposure to drugs and other environmental factors, can significantly influence the microbiome [3,4]. Alterations in microbial composition can affect physiological processes related to the development of a wide range of diseases [5,6]. Dysbiosis of the fecal, salivary, or skin microbiome occurs in patients with inflammatory, autoimmune [7,8,9,10,11,12,13,14,15,16,17,18,19,20,21], and cutaneous diseases [22,23] compared to healthy controls (HCs).

Behçet’s disease (BD), a rare systemic inflammatory disorder most prevalent in Eastern and Central Asian and Eastern Mediterranean countries [24], affects multiple sites in the body and produces mucocutaneous and systemic lesions. The most common mucocutaneous lesions are recurrent oral aphthous ulcers, recurrent genital ulcers, and other cutaneous manifestations, such as pustular vasculitis lesions, erythema nodosum, and Sweet’s syndrome-like lesions, with a painful sensation often reported. Systemic lesions are usually characterized by ocular lesions, especially posterior uveitis, arthritis, and systemic vascular, central nervous system, and gastrointestinal involvement. Most BD-related lesions have a tendency toward chronic progression, with acute flare-ups at any given time [6,25]. Although various factors related to the pathogenesis of BD, such as genetics [26,27,28,29,30,31], and infectious [32,33], immunologic [34,35,36], and epigenetic [37] factors, have been actively investigated, the etiologies and pathogenic factors related to BD have not yet been elucidated.

In studies analyzing fecal or salivary microbial differences between patients with BD and HCs, significantly different microbial diversities have mostly been noted between the two groups. Additionally, several specific microbiomes of patients with BD have been found to be either more or less abundant than those of their HCs [10,11,16,17]. Although the results have been heterogeneous by region and study design, it appears that these microbial alterations may play an important role in immune aberration and the triggering of inflammatory processes in the pathogenesis of BD.

To the best of our knowledge, there have been no studies in which the factors that influence the microbiome are considered as covariates, and in which fecal and salivary samples are extracted from the same patients. Additionally, it remains unclear as to how different alterations in microbiome composition are related to disease activity in patients with BD.

In this study, we hypothesized that the fecal and salivary microbiomes are altered in BD according to disease activity, and that this microbial dysbiosis is associated with the pathophysiology of BD. Therefore, we investigated the fecal and salivary microbiomes of patients with BD, patients with recurrent aphthous ulcers (RAUs) as disease controls, and matched HCs selected from among the family members of the patients. We also compared the differences in the taxonomic composition of the microbiota in these groups, and investigated whether the changes in the microbiome composition of patients with BD were dependent on the activity of their disease.

## 2. Materials and Methods

### 2.1. Study Participants

Patients diagnosed with BD at Ajou University Hospital between January 2018 and May 2021 were enrolled in this study. Diagnosis was confirmed using the international criteria of the International Study Group on BD [38] and the diagnostic criteria of the BD Research Committee of Japan (2003) [39]. We excluded patients who were younger than 19 years at the initial visit to our clinic; who had previous metabolic diseases, such as diabetes, thyroid diseases, and hypothalamic diseases, such as liver cirrhosis and chronic xerostomia; who had bowel inflammation, bowel tumors, periodontal pockets >4 mM, cavitated carious lesions, oral abscesses, oral tumors, oral candidiasis, loss of teeth, or bleeding on probing of more than 10% at the time of sampling; who had used probiotics or antibiotics within six months of study enrollment; who had any history of dietary restrictions. Patients diagnosed with RAU were assigned to the disease control group. For the HC group, we selected from among the family members of each patient a healthy person who had been eating one or more meals per day with the patient and did not meet the exclusion criteria listed above.

Nine patients with BD, seven with RAU, nine BD-matched HCs, and seven RAU-matched HCs were enrolled in the study. At the initial visit, all patients were in the active phase of the disease, and laboratory tests, including examination of human leukocyte antigen (HLA)-B51 levels and levels of inflammatory markers, such as the erythrocyte sedimentation rate (ESR) and C-reactive protein (CRP), were performed to evaluate disease severity. After an average of 173.3 days, seven patients with active BD converted to the disease-inactive phase, which was defined as the period in which patients exhibited no mucocutaneous or systemic lesions and no other BD-related findings, and had inflammatory marker levels following systemic treatment lower than those at the initial visit. We collected and analyzed fecal and salivary samples from those seven patients with BD during the active period of the disease and again in the inactive phase of BD.

### 2.2. Fecal and Salivary Sampling

Fecal and salivary samples were collected using OMNIGENE^®^-GUT and OMNIGENE^®^-ORAL kits (DNA Genotek Inc., Ottawa, ON, Canada) by the study participants. A spatula was used to collect a small amount of fecal sample, which was transferred to the top of the collection tube. The sample was then leveled by scraping horizontally across the top of the tube, and the tube was capped tightly. The sealed tube was shaken for a minimum of 30 s. Saliva without bubbles was collected until the fill line was reached, and the funnel lid was closed tightly. The funnel was then removed, and the tube was tightly closed. Finally, the sealed tube was shaken for 10 s. All samples were submitted for analysis as soon as they were collected.

### 2.3. DNA Extraction, Polymerase Chain Reaction Amplification, and Sequencing

Full genetic analyses of the microbiomes in all the submitted fecal and salivary samples were performed through DNA extraction, polymerase chain reaction (PCR) amplification, and sequencing. Total DNA from each sample was extracted using FastDNA^®^ SPIN Kits for Soil (MP Biomedicals, CA, USA), in accordance with the manufacturer’s instructions.

PCR amplification of the extracted DNA was performed using fusion primers targeting regions V3 to V4 of the 16S ribosomal RNA gene. For bacterial amplification, the fusion primers 341F and 805R were used (Table 1). Amplification was carried out under the following conditions: initial denaturation at 95 °C for 3 min, followed by 25 cycles of denaturation at 95 °C for 30 s, primer annealing at 55 °C for 30 s, and extension at 72 °C for 30 s, with a final elongation at 72 °C for 5 min. The PCR products were confirmed using 1% agarose gel electrophoresis and visualized using a GelDoc system (Bio-Rad, Hercules, CA, USA). The amplified products were purified using CleanPCR (CleanNA, Alphen aan den Rijn, The Netherlands). Equal concentrations of the purified products were pooled, and non-target short fragments were eliminated using CleanPCR (CleanNA, Alphen aan den Rijin, The Netherlands). Product quality and size were assessed on a Bioanalyzer 2100 (Agilent, Palo Alto, CA, USA) using a DNA 7500 chip.

Mixed amplicons were pooled, and sequencing was carried out at ChunLab, Inc. (Seoul, Korea) using the Illumina MiSeq Sequencing system (Illumina, Inc., CA, USA), according to the manufacturer’s instructions.

### 2.4. Sequence Analysis

Processing of the raw reads started with quality checking and the filtering of low-quality reads using Trimmomatic ver. 0.32 [40]. Operational taxonomic units were obtained, and diversity calculations and biomarker discovery were performed using in-house programs from ChunLab, Inc. (Seoul, Korea). The alpha diversity, which reflects the diversity within a community, was evaluated using the Chao1 and Shannon diversity indices. To visualize sample differences, beta diversity, which reflects the diversity between communities, was calculated using Bray-Curtis principal coordinate analysis (PCoA) plots in a three-dimensional space. Taxonomic biomarkers were identified using the linear discriminant analysis effect size (LEfSe) algorithm [41], which is a statistical comparison algorithm.

### 2.5. Statistical Analysis

We compared microbial alpha diversity using the Wilcoxon rank-sum test. The significant *p*-values of the difference in beta diversity between the two groups were evaluated using permutational multivariate analysis of variance (PERMANOVA). After comparing the relative abundance of each taxon using the Wilcoxon rank-sum test, the microbial taxa showing significant differences were used for linear discriminant analysis (LDA) to obtain the effect size. We included all taxa with an LDA score of more than two in the taxonomic biomarker analysis plots. However, we set the cut-off value of the LDA score to 4 because the taxa with an LDA score of less than 4 in the LEfSe algorithm displayed minimal differences in microbiome abundance between the groups compared in this study. All results were considered statistically significant when the two-tailed *p*-value was < 0.05.

### 2.6. Ethics Approval

This study was approved by the Ajou Institutional Review Board (IRB No.: AJIRB-BMR-MDB-15-341) prior to commencement and was performed in accordance with the ethical standards laid down in the 1964 Declaration of Helsinki and its later amendments, or comparable ethical standards.

## 3. Results

### 3.1. Subject Characteristics

The demographic and clinical characteristics of the study participants are presented in Table 2. The median disease durations of patients with BD or RAU were 5.0 (1.5–9.0) and 1.0 (0.5–2.0) years, respectively. There was no family history of disease in the BD and RAU groups. The evaluation of disease characteristics showed that all patients with BD had oral aphthous ulcers, genital ulcers, and skin lesions; however, only two out of seven patients with BD had underlying uveitis. All patients with RAU had oral aphthous ulcers but none had genital ulcers, skin lesions, or ocular lesions. None of the patients with BD or RAU had a history of arthritis or gastrointestinal or central nervous system lesions. The median levels of inflammatory markers, such as the ESR and CRP, were higher in patients with BD than in those with RAU. All patients with BD were HLA-B51-positive, but only one out of seven with RAU was HLA-B51-positive. All patients with BD and RAU had been treated with systemic steroids, and most patients with BD (88.9%) had been treated with colchicine. Only two patients from each group had used probiotics six months before enrollment, and none had used antibiotics before the study.

### 3.2. Alpha Diversity of the Fecal Microbiome in Patients with Active BD Was Significantly Different from That of Unmatched HCs but Not Matched HCs or Patients with Active RAUs

We determined the distances between the taxonomic compositions of the microbiomes of the different groups using alpha diversity (Figure 1) and beta diversity (Figure 2). In the fecal and salivary sample analyses, there were no significant differences in alpha diversity between the active BD group and the BD-matched HC group. In addition, the alpha diversities of the fecal and salivary microbiomes in the active RAU group were not significantly different from those in the RAU-matched HC group. However, the alpha diversities of the fecal samples in the RAU-matched HC group were significantly different from those in the active BD group (*p* = 0.03), and statistically significant differences between the RAU-matched HC and BD-matched HC groups were also observed (*p* = 0.03). By comparing the taxa in the microbiomes of the active BD group with those of the active RAU group, the alpha diversities of the fecal and salivary samples were found not to be significantly different between the two groups. Beta diversity analysis found no significant differences between all groups, but PCoA on the Bray–Curtis distance showed distinguishable clustering of the active BD group compared with the BD-matched HC group. 

### 3.3. Fecal Bacteroides Uniformis and Salivary Rothia Mucilaginosa Levels Were Higher in Patients with BD Than in BD-Matched HCs and Patients with RAUs, Respectively

The average taxonomic composition at the phylum level is shown in Figure 3. In all groups, *Bacteroidetes* and *Firmicutes* were the most prevalent taxa in the stool, and the phyla *Firmicutes*, *Proteobacteria*, and *Bacteroidetes* were the most prevalent taxa in the saliva.

We compared the relative abundances of taxa in the microbiomes of the disease groups with those in the matched HC group. In fecal sample analyses, among the taxa showing significantly different relative abundances between the active BD and BD-matched HC groups, *B. uniformis* was the most abundant taxon in the active BD group. Its abundance was significantly higher than that in the BD-matched HC group (*p* = 0.038) (Figure 4A). In salivary sample analyses, the active BD group showed a significantly higher abundance of *Lachnoanaerobaculum* than the BD-matched HC group (*p* = 0.038), but the difference was very small (Figure 4B). The relative abundance of salivary *Rothia mucilaginosa* in the active BD group was significantly higher than that in the active RAU group (*p* = 0.01) (Figure 5B). However, the differences between the groups in terms of the relative abundance of the fecal microbiomes were minor (Figure 5A).

We also compared the relative abundance of taxa in the RAU group with the RAU-matched HC group. There was a minor difference in the relative abundances of fecal and salivary taxa between the two groups (data not shown).

### 3.4. Alpha Diversity of the Fecal Microbiome Decreased in Patients with BD When the Disease Changed from the Active to Inactive Phase

We compared paired samples obtained from patients with BD in the active phase and in the inactive phase. The alpha diversities of the microbiomes in the stool of patients with inactive BD were significantly lower than those in patients with active BD (*p* = 0.035 and *p* = 0.048, respectively) (Figure 6), while no significant differences in beta diversity between groups were observed (Figure 7). In the relative abundance analyses comparing the three groups at once (Figure 8), the abundance of fecal *B. uniformis* was significantly different between the active BD, inactive BD, and BD-matched HC groups (*p* = 0.042), and was highest in the active BD group. The abundances of order *Actinomycetales* and family *Actinomycetaceae* in saliva were significantly higher in the active BD group, and there was a minimal difference in the relative abundances of these taxa between groups. However, significantly different abundances of microbes were not detected in the fecal microbiomes of the BD-matched HC group and the salivary microbiomes of the inactive BD group.

### 3.5. The Relative Compositions in the Fecal and Salivary Microbiomes Were Different in Patients with BD According to Their Having Uveitis or Not 

We performed subgroup analyses of the differences in taxa in the microbiomes according to the presence of underlying uveitis in patients with active BD. The alpha and beta diversities were not significantly different between the groups, yet, the abundances of various bacteria in the fecal and salivary samples from both groups were significantly different. The abundances of fecal *Faecalibacterium prausnitzii group* and *Bifidobacterium adolescentis group* and salivary *Streptococcus pneumoniae group*, *Streptococcus peroris group*, and *Neisseria sicca group* in those with uveitis were significantly higher than in those without uveitis. However, the abundances of fecal genus *Clostridium_g24* and salivary genus *Veillonella* in those with uveitis were significantly lower than in those without uveitis (Figure 9).

## 4. Discussion

In this study, we investigated alterations in the microbiome of patients with BD by comparing the microbiomes of patients with BD with those of three control groups: namely, BD-matched HCs who shared a home and at least one meal per day with patients with BD; patients with RAU with symptoms similar to those with BD; RAU-matched HCs as unrelated HCs. The alpha diversities of the fecal microbiomes in patients with active BD did not significantly differ from those in BD-matched HCs or patients with RAUs, although these were significantly different from those in unrelated HCs. However, fecal microbiome alpha diversities in patients with active BD were significantly reduced after the disease was shifted from an active to an inactive state. The average taxonomic composition of phylum *Bacteroidetes* was the most prevalent in the stool of both the BD patients and their-matched healthy subjects, which is consistent with previous findings [42]. Our results also demonstrated that the abundance of fecal bacterial species—*B. uniformis*—was altered in patients with active BD compared to that in the control groups and according to disease activity. The relative abundances of several taxa in the fecal and salivary microbiomes differed between patients with BD with uveitis and those without. Unlike in the fecal microbiomes, the diversity and composition of the salivary microbiomes in patients with BD were not significantly different from those in the control groups or according to disease activity.

We found that the alpha diversities of the fecal microbiomes in patients with active BD were similar to those in matched HCs, but different from those in unmatched HCs. A previous study found that patients with BD showed differences in fecal bacterial alpha diversity compared to unmatched normal healthy subjects [11]. Recently, van der Houwen et al. reported that the alpha diversities were significant different between patients with BD and HCs, but this difference may be more strongly driven by factors such as diet than by disease status [43]. In contrast to a study by Coit et al. [10], our results showed no differences in salivary microbiome diversity between patients with BD and controls. One possible explanation to the discrepancy between these results may be attributed to the differences in the background of the HC subjects.

Our findings demonstrated an association between fecal microbiome and BD. We observed (1) an alteration in the alpha diversity of the fecal microbiome in patients with BD according to disease activity, (2) an increased *B. uniformis* abundance in patients with active BD compared to that in matched HCs or those with an inactive disease state, and (3) a tendency toward clustering in fecal microbiome beta diversity analysis (Figure 2B), despite the latter not being statistically significant, possibly due to the small sample size. One previous study revealed that BD disease activity parameters are not correlated with any fecal taxa prevalent in BD [17], and other studies regarding patients with BD [43] or ulcerative colitis [44] indicated that specific microbiomes are associated with disease severity. However, unlike our study, these previous studies assessed the association between disease severity and microbial composition using only the data from the initial visit of each patient. Conversely, in our study, we analyzed the fecal and salivary samples of all enrolled patients with BD at the first visit and when they were in the inactive disease phase, and assessed whether changes in the composition of their microbiomes were dependent on their disease activity. Therefore, our findings may be more clinically relevant, since we observed microbial changes according to disease severity in the same patients with BD. Notably, we should mention that during the treatment period, most patients with BD stated that they adjusted their diet, such as reducing instant food and consuming a fiber-rich diet. We did not include the BD-matched HC sample at the time point of the inactive phase of BD, which is one limitation of this study. A fiber-rich diet is associated with an increase in short-chain fatty acid (SCFA) producers and non-SCFA *B. uniformis* in the gut microbiome [45]. However, an alteration in the abundance of SCFA producers during BD disease activity shift was not observed, but a decrease in *B. uniformis* abundance was observed. Moreover, the abundance of *B. uniformis* was higher in patients with active BD than in those with diet-matched HCs. Therefore, a diet change might not have been the primary factor altering the abundance of *B. uniformis* in the gut microbiota during BD disease activity shift. Given our previous observation that SCFA negatively affects the production of inflammatory cytokines in patients with BD [46], we speculated an alteration in the abundance of SCFA producers during BD disease activity shift. Our findings do not concord with this speculation, but rather show an alteration in *B. uniformis* abundance, suggesting a possible role of the gut microbiome in BD pathogenesis. Further investigations regarding the role of *B. uniformis* in immunological and metabolic regulation are required. 

This study also showed that salivary *Rothia mucilaginosa* was more abundant in the active BD group than in the active RAU group. A previous study investigating the salivary and oral mucosal microbial communities in BD showed a noticeable shift toward *R. mucilaginosa* from the ulcerated mucosa of orally active BD to the non-ulcerated mucosa of orally active BD [16]. The reason for increased salivary *R. mucilaginosa* is not currently known. Unlike RAU, *R. mucilaginosa* may be associated with localized progression and systemic manifestations of BD. RAU is a relatively localized disease confined to the oral cavity, whereas BD is characterized not only by the most common and initially recurrent oral aphthous, but also by later extraoral inflammation [6].

In the subgroup analyses of samples with underlying uveitis, the abundances of various bacteria in the fecal and salivary samples from both groups were significantly different. Among them, the abundance of *B. adolescentis* in the fecal samples of patients with active BD with uveitis was significantly higher than that in those without uveitis. Microbial dysbiosis is involved in the pathogenesis of uveitis through autoimmunity, and differences in the composition of the microbiota have been reported in patients with uveitis [47,48,49]. Kalyana et al. [49] reported that levels of *B. adolescentis* decreased by several folds in idiopathic uveitis, an observation that is not in accordance with our results. However, Kalyana et al.’s study participants were diagnosed with localized uveitis without systemic involvement. Given that this bacterium is also a butyrate non-producer [45], it can be speculated that the effects of *B. adolescentis* on uveitis in systemic disease may be different from those in localized disease. Further studies are required to reconcile the differences in our results and those of Kalyana et al.

This study has some limitations. Our sample size was small owing to the very low incidence of BD. Moreover, as mentioned above, the lack of inclusion of the BD-matched HC sample at the time point of the inactive phase of BD is another limitation. However, this study included dietary habit-matched family members of each patient as HCs to minimize the dietary effect on the microbial compositions and analysis of gut microbiomes in BD patients according to disease activity. 

## 5. Conclusions

This study revealed alterations in the fecal microbiome in patients with BD according to disease activity, and an association of the abundance of fecal bacterial species with BD disease activity and uveitis symptoms. Further animal model studies or immunologic studies are warranted to address whether gut microbial dysbiosis may contribute to BD pathogenesis, and whether the modulation of gut microbiomes may be a new therapeutic target against BD. 

## Figures and Tables

**Figure 1 microorganisms-09-01449-f001:**
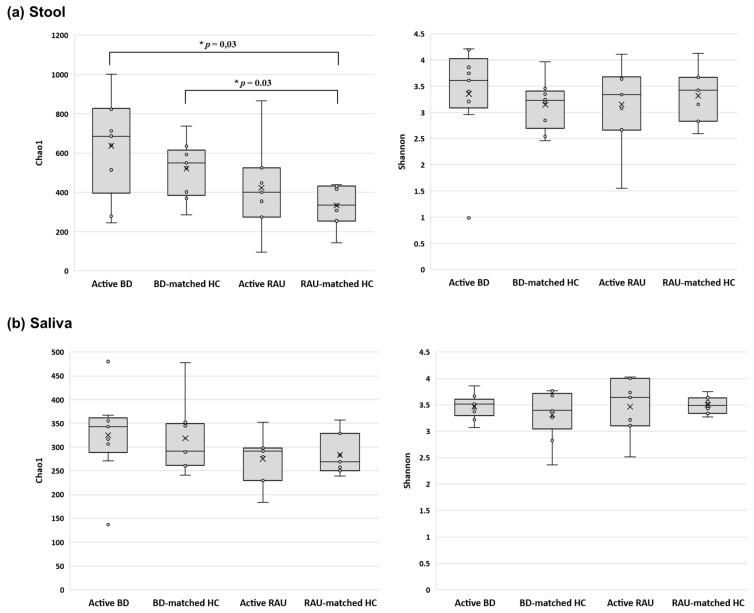
Comparison of the taxonomic alpha diversities of the microbiomes of active BD, BD-matched HC, active RAU, and RAU-matched HC groups. The alpha diversity indices calculated using the Chao1 and Shannon diversity indices in stool (**a**) and saliva (**b**) of the active BD group were higher than those in the BD-matched HC group; however, the difference was not statistically significant. The alpha diversity indices in the stool and saliva samples of the active RAU group were not significantly different from those of the RAU-matched HC group. On comparing the taxa in the microbiomes of patients with active BD with those of patients with active RAU, as a disease control, the alpha diversities of fecal and salivary samples were not significantly different between the two groups. However, the taxonomic richness of the microbiome (Chao1 index) in stool of the active BD and BD-matched HC groups was significantly higher than that of the RAU-matched HC group (*p* = 0.03 and *p* = 0.03, respectively). BD, Behçet’s disease; RAU, recurrent aphthous ulcer; HC, healthy control.

**Figure 2 microorganisms-09-01449-f002:**
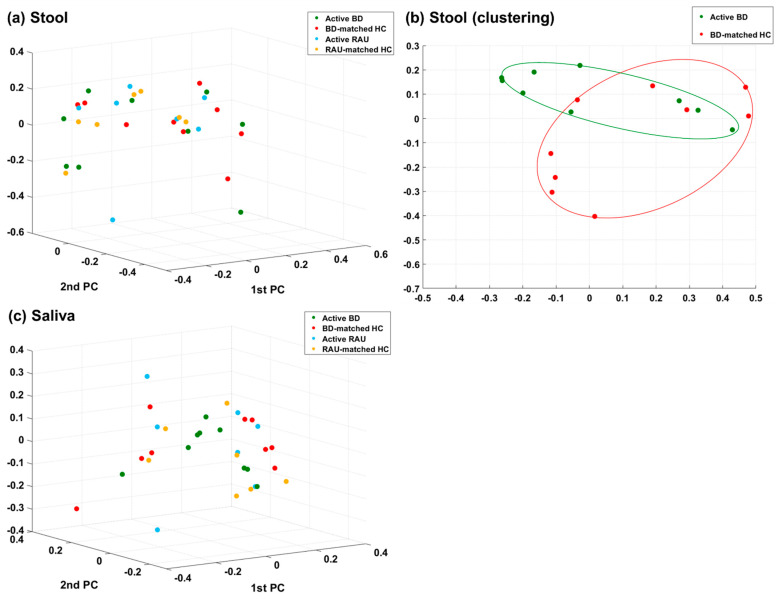
Comparison of the taxonomic beta diversities of the microbiome of the active BD (green), BD-matched HC (red), active RAU (blue), and RAU-matched HC (yellow) groups. Differences in beta diversity between the groups were visualized using Bray–Curtis principal coordinate analysis (PCoA) plots in a three-dimensional space. The significant *p*-values of the differences in beta diversity between the groups were evaluated using permutational multivariate analysis of variance (PERMANOVA). In stool (**a**) and saliva (**b**), there were no significant differences in the beta diversity between groups. However, a distinguishable clustering of the active BD group compared with the BD-matched HC group was noted in stool (**c**). BD, Behçet’s disease; RAU, recurrent aphthous ulcer; HC, healthy control.

**Figure 3 microorganisms-09-01449-f003:**
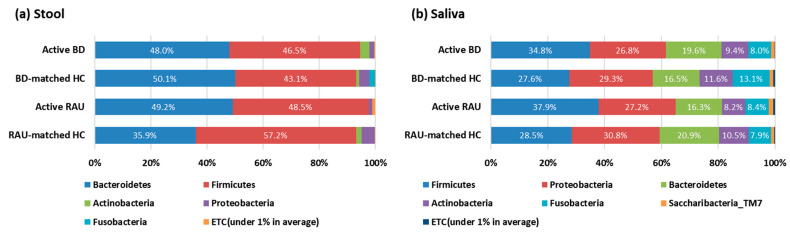
Average taxonomic compositions of the microbiomes (phylum level). In stool (**a**), the phyla *Bacteroidetes* and *Firmicutes* were the most frequent taxa in the active BD, BD-matched HC, active RAU, and RAU-matched HC groups. In saliva (**b**), the phyla *Firmicutes*, *Proteobacteria*, *Bacteroidetes*, *Actinobacteria*, and *Fusobacteria* were the most prevalent taxa in all groups. BD, Behçet’s disease; RAU, recurrent aphthous ulcer; HC, healthy control.

**Figure 4 microorganisms-09-01449-f004:**
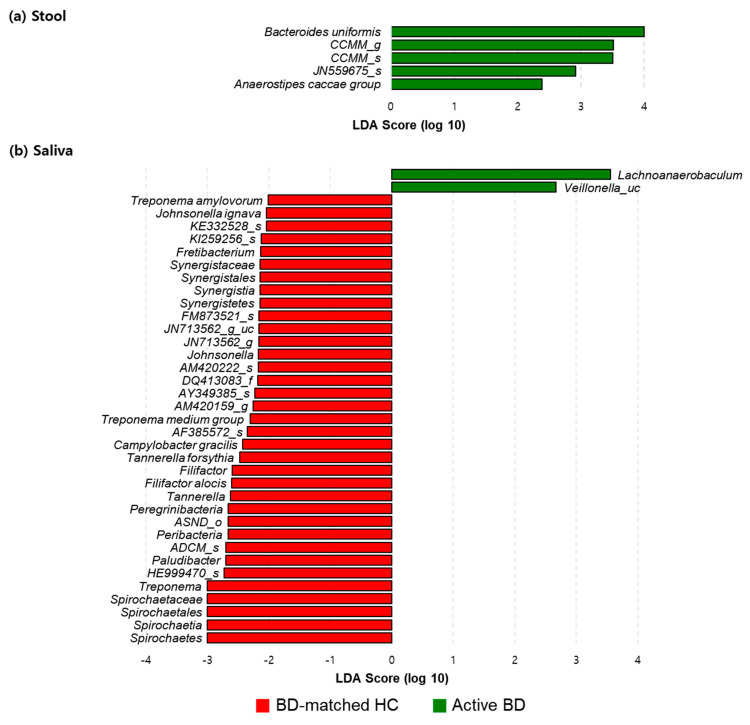
Relative abundances of taxa in the microbiomes of stool (**a**) and saliva (**b**) in active BD (green) and BD-matched HC (red) groups. The linear discriminant analysis (LDA) scores (log 10) were calculated using the LDA Effect Size (LEfSe) algorithm. Taxa with an LDA score (log 10) of more than 4 were *Bacteroides uniformis* in the stool of active BD (*p* = 0.038).

**Figure 5 microorganisms-09-01449-f005:**
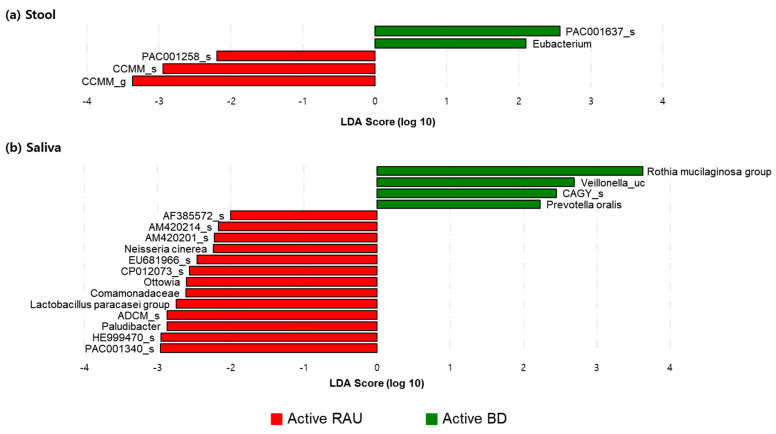
Relative abundances of taxa in the microbiomes of stool (**a**) and saliva (**b**) in active BD (green) and active RAU (red) groups. The linear discriminant analysis (LDA) scores (log 10) were calculated using the LDA Effect Size (LEfSe) algorithm. Salivary *Rothia mucilaginosa* was significantly higher in active BD than in active RAU (*p* = 0.01), although the taxon with an LDA score (log 10) of more than 4 was not detected in the fecal and salivary samples.

**Figure 6 microorganisms-09-01449-f006:**
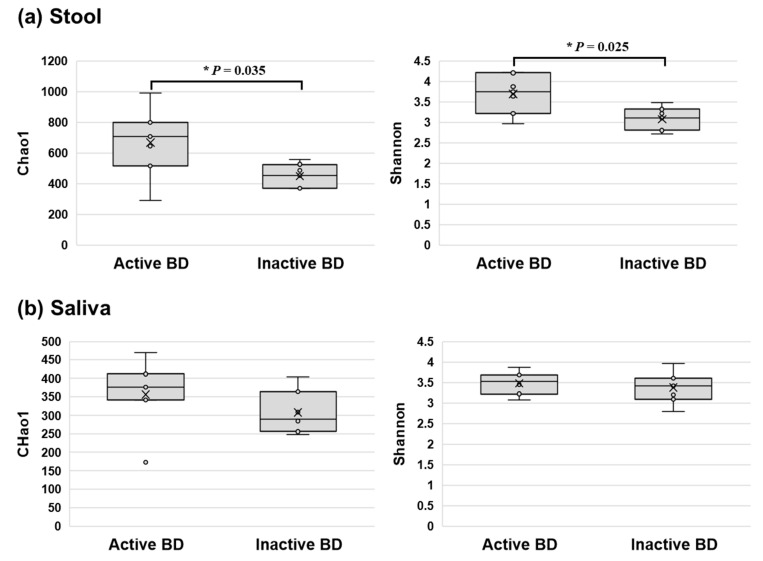
Comparison of the taxonomic alpha diversities of microbiomes in the active and inactive BD groups. The alpha diversity scores calculated using the Chao1 and Shannon diversity indices in the stool (**a**) of the active BD group were significantly higher than those of the inactive BD group (*p* = 0.035 and *p* = 0.025, respectively). However, in salivary samples (**b**), no difference in alpha diversity indices was observed between the two groups. BD, Behçet’s disease.

**Figure 7 microorganisms-09-01449-f007:**
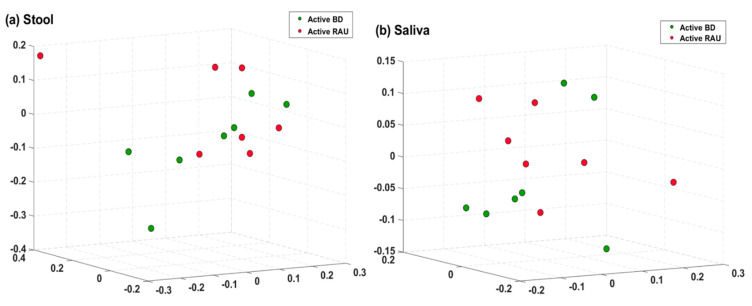
Comparison of the taxonomic beta diversities of microbiomes of the active BD (green) and inactive BD (green) groups. Differences in beta diversity between the groups were visualized using Bray-Curtis principal coordinate analysis (PCoA) plots in a three-dimensional space. The significant p-values of the differences in beta diversity between groups were evaluated using permutational multivariate analysis of variance (PERMANOVA). In stool (**a**) and saliva (**b**), there were no significant differences in the beta diversity between groups. BD, Behçet’s disease.

**Figure 8 microorganisms-09-01449-f008:**
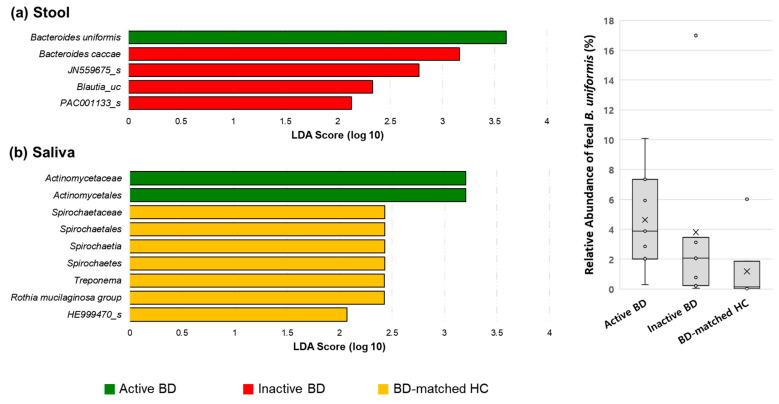
Relative abundances of taxonomic compositions of microbiomes of stool (**a**) and saliva (**b**) in active BD (green), inactive BD (red), and BD-matched HC (yellow) groups. The linear discriminant analysis (LDA) scores (log 10) were calculated using the LDA Effect Size (LEfSe) algorithm by comparing these three groups at once. The abundance of fecal *Bacteroides uniformis* was significantly different between the active BD, inactive BD, and BD-matched HC groups (*p* = 0.042), and was the highest in the active BD group. However, a minor difference in the relative abundances in the salivary microbiomes between groups was observed, and significantly different abundances of microbes were not detected in the fecal microbiomes of the BD-matched HC and the salivary microbiomes of the inactive BD group. BD, Behçet’s disease; HC, healthy control.

**Figure 9 microorganisms-09-01449-f009:**
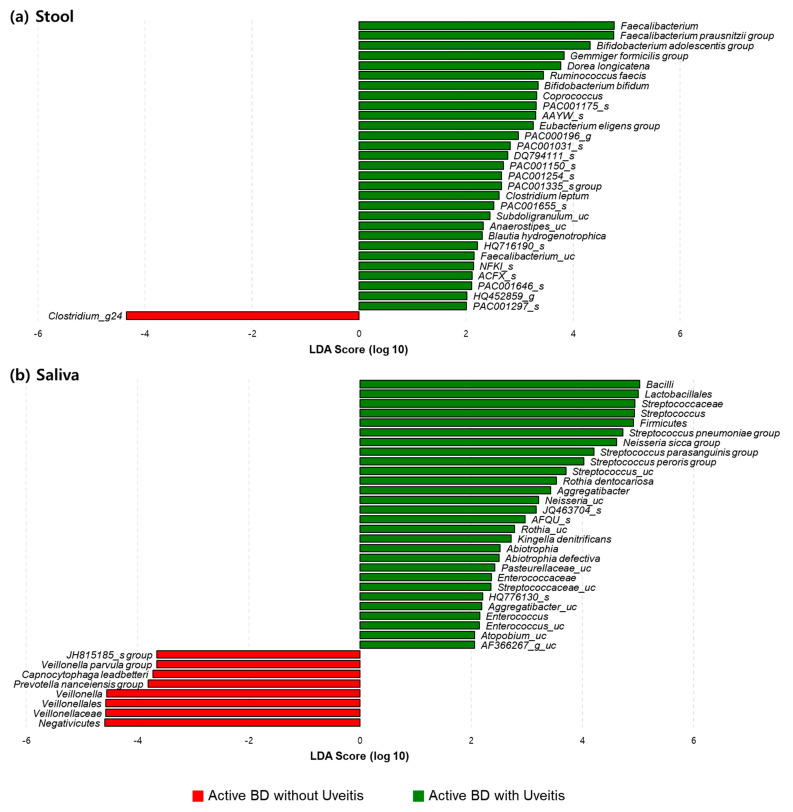
Relative abundances of taxonomic compositions of microbiomes of stool (**a**) and saliva (**b**) in active BD with uveitis (green) and active BD without uveitis (red) groups. The linear discriminant analysis (LDA) scores (log 10) were calculated using the LDA Effect Size (LEfSe) algorithm. The various taxa showed a LDA score (log 10) of more than 4 in the stool and saliva of both groups. BD, Behçet’s disease.

**Table 1 microorganisms-09-01449-t001:** Fusion primers used for bacterial amplification.

Fusion Primers	
341F	5′-AATGATACGGCGACCACCGAGATCTACAC-XXXXXXX-TCGTCGGCAGCGTC-AGATGTGTATAAGAGACAG-CCTACGGGNGGCWGCAG-3′
805R	5′- CAAGCAGAAGACGGCATACGAGAT-XXXXXXXX-GTCTCGTGGGCTCGG-AGATGTGTATAAGAGACAG-GACTACHVGGGTATCTAATCC-3′

The underlined sequences indicate the target-region primers.

**Table 2 microorganisms-09-01449-t002:** Clinical characteristics of the study population.

	Active BD (*n* = 9)	BD-Matched HC (*n* = 9)	Active RAU (*n* = 7)	RAU-Matched HC (*n* = 7)
**Age, median (IQR), years**	33 (28.5–45.5)	53 (33–56.5)	47 (40–66)	44 (36–67)
**Sex**				
Male, *n* (%)	1 (11.1)	4 (44.4)	2 (28.6)	3 (42.9)
Female, *n* (%)	8 (88.9)	5 (55.6)	5 (71.4)	4 (57.1)
**Alcohol history, *n* (%)**	2 (22.2)	5 (55.6)	1 (14.3)	4 (57.1)
**Smoking**				
Smokers, *n* (%)	2 (22.2)	0 (0)	0 (0)	0 (0)
Ex-smokers, *n* (%)	1 (11.1)	1 (11.1)	2 (28.6)	2 (28.6)
Nonsmokers, *n* (%)	6 (66.7)	8 (88.9)	5 (71.4)	0 (0)
**Family history of each disease, *n* (%)**	0 (0)	N/A	0 (0)	N/A
**Disease duration, median (IQR), years**	5.0 (1.5–9.0)	N/A	1.0 (0.5–2.0)	N/A
**Disease characteristics**				
Oral aphthous ulcers, *n* (%)	9(100)	0 (0)	7 (100)	0 (0)
Genital ulcers, *n* (%)	9 (100)	0 (0)	0 (0)	0 (0)
Skin lesions, *n* (%)	9 (100)	0 (0)	0 (0)	0 (0)
Uveitis, *n* (%)	2 (22.2)	0 (0)	0 (0)	0 (0)
Arthritis, *n* (%)	0 (0)	0 (0)	0 (0)	0 (0)
GI lesions, *n* (%)	0 (0)	0 (0)	0 (0)	0 (0)
CNS lesions, *n* (%)	0 (0)	0 (0)	0 (0)	0 (0)
ESR, median (IQR), mm/h	22.0 (13.5–26.0)	N/A	9.0 (2.0–37.0)	N/A
CRP, median (IQR), mg/dL	0.27 (0.07–1.15)	N/A	0.12 (0.03–0.30)	N/A
**HLA-B51**				
Positive, *n* (%)	9 (100)	N/A	1 (14.3)	N/A
Negative, *n* (%)	0 (100)	N/A	6 (85.7)	N/A
**Previous treatment**				
Systemic steroids, *n* (%)	9 (100)	N/A	7 (100)	N/A
Colchicine, *n* (%)	8 (88.9)	N/A	2 (28.6)	N/A
Azathioprine, *n* (%)	1 (14.3)	N/A	0 (0)	N/A
Cyclosporine, *n* (%)	1 (14.3)	N/A	0 (0)	N/A
Antibiotic, *n* (%)	0 (0)	0 (0)	0 (0)	0 (0)
**Previous probiotic use, *n* (%)**	2 (22.2)	2 (22.2)	2 (28.6)	2 (28.6)

BD, Behçet’s disease; RAU, recurrent aphthous ulcer; HC, healthy control; GI, gastrointestinal; CNS, central nervous system; ESR, erythrocyte sedimentation rate; CRP, C-reactive protein; HLA, human leukocyte antigen; IQR, interquartile range; N/A, not applicable.

## Data Availability

Metagenomic sequencing data for all BD, RAU, and HC samples were deposited in NCBI with the accession number PRJNA677855.

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
