# Peer review of "Alteration of the Fecal but Not Salivary Microbiome in Patients with Behçet’s Disease According to Disease Activity Shift"

_microorganisms, 2021, doi:10.3390/microorganisms9071449_

Round 1
Reviewer 1 Report
In the current study the author provided valuable data on influence of microbial composition on disease severity and recovery of BD patients compared to those of recurrent aphthous ulcers (RAUs) patients and to dietary habit-matched healthy controls (HCs). Hence, it is possible that increased abundance of microbes leading to BD recovery may provide deeper insight into further investigations on microbial interaction or effect of secondary metabolites from these microbes on disease condition through manipulation of cell signaling. Moreover, targeting the increased microbes in the pathogenesis of BD may provide future therapeutic approach for BD. This article is well designed, clear and well written.
Point-by-points comments:
1. High resolution should be provided for figure 2 and 7. The font size in x- and y-axis legends in these figures should be increased.
2. Author should include family history N (%) in characteristic criteria in the study populations for BD and RAUs, as studies evidenced that majority of patients with RAUs have a family history of this disease.
3. Including data from microbial abundance of healthy subject will provide more clear comparison on microbial alteration between healthy, inactive and active BD patients during BD disease activity.
4. Zi ye et al, “metagenomic study of the gut microbiome in Behcet’s disease“, https://www.ncbi.nlm.nih.gov/pmc/articles/PMC6091101/, here they found that Bacteroidetes, are dominant phyla in both BD patients and healthy controls. In the current study, Relative abundance of Bacteriodetes uniformis was significantly higher in the active BD group than inactive or BD-matched HC groups. Author should cite the above article and discuss this part.
5. In discussion section, author states that “the abundance of B. uniformis, which is not a butyrate producer, could affect the composition of other butyrate-producing microbiomes, resulting in inflammation." It is a general speculation on basis of microbial dysbiosis.
Did author compared the abundance of B. uniformis from active BD with inactive ones (patients after recovered from BD). Including fiber-rich diet in BD patients during recovery period and further comparison on the alteration in B. uniformis, in both active and inactive BD will provide a clear conclusion on alteration in butyrate-producing bacteria may be a major factor involved in inflammation and pathogenesis of BD.
Author Response
Response to Reviewer 1 Comments
Point 1: High resolution should be provided for figure 2 and 7. The font size in x- and y-axis legends in these figures should be increased.
Response 1: Thank you for your suggestion. We revised Figures 2 (page 7) and 7 (page 11) accordingly.
Point 2: Author should include family history N (%) in characteristic criteria in the study populations for BD and RAUs, as studies evidenced that majority of patients with RAUs have a family history of this disease.
Response 2: Thank you for your suggestion, we have added the family history N (%) of each disease to Table 2 and revised the text accordingly (Results section page 4, lines 157-158). We found no family history of disease in either the BD or RAU group.
Point 3: Including data from microbial abundance of healthy subject will provide more clear comparison on microbial alteration between healthy, inactive and active BD patients during BD disease activity.
Response 3: As suggested, the relative abundance of inactive and active BD patients and their matched healthy subjects have been shown in the Figure 8. For clear comparison, we have added the extra panel to Figure 8 showing the compositional change of the fecal B. uniformis during BD disease activity shift (page 12).
Point 4: Zi ye et al, “metagenomic study of the gut microbiome in Behcet’s disease “, https://www.ncbi.nlm.nih.gov/pmc/articles/PMC6091101/, here they found that Bacteroidetes, are dominant phyla in both BD patients and healthy controls. In the current study, Relative abundance of Bacteriodetes uniformis was significantly higher in the active BD group than inactive or BD-matched HC groups. Author should cite the above article and discuss this part.
Response 4: In our study, phylum Bacteroidetes was the most prevalent taxon in the stool of both BD patients and healthy controls, consistent with previous findings (Zi ye et al., 2018). Among the bacterial species that belong to phylum Bacteroidetes, the abundance of only Baceroides uniformis was significantly different between the active BD and their matched HC groups. As suggested, we have cited the previous article and discussed the findings in the revised Discussion section (page 14, lines 306-308).
Point 5: In discussion section, author states that “the abundance of B. uniformis, which is not a butyrate producer, could affect the composition of other butyrate-producing microbiomes, resulting in inflammation." It is a general speculation on basis of microbial dysbiosis.
Response 5: Thank you for your comment and we agree with your assessment. Accordingly, we revised the text as follows (page 14, lines 352-358):
“Given our previous observation that SCFA negatively affects the production of inflammatory cytokines in patients with BD (Yun et al., Ann Dermatol 2018), we speculated an alteration in the abundance of SCFA producers during BD disease activity shift. Our findings do not concord with this speculation but show the alteration in B. uniformis abundance, suggesting a possible role of the gut microbiome in BD pathogenesis. Further investigations regarding the role of B. uniformis in immunological and metabolic regulation are required.”
Point 6: Did author compared the abundance of B. uniformis from active BD with inactive ones (patients after recovered from BD). Including fiber-rich diet in BD patients during recovery period and further comparison on the alteration in B. uniformis, in both active and inactive BD will provide a clear conclusion on alteration in butyrate-producing bacteria may be a major factor involved in inflammation and pathogenesis of BD.
Response 6: Thank you for your comment. Accordingly, we have added additional information regarding patients’ diet and discussed the possible effect of fiber rich diet on gut microbial composition in the revised manuscript. (14, lines 342-352).
“Notably, during the treatment period, most patients with BD adjusted their diet, such as reducing instant food and consuming a fiber-rich diet. We did not include the BD-matched HC sample at the time point of the inactive phase of BD, which is one limitation of this study. A fiber-rich diet is associated with an increase in short-chain fatty acid producers and non-SCFA B. uniformis in the gut microbiome (Louis et al., Environ Microbiol. 2017). However, an alteration in the abundance of SCFA producers during BD disease activity shift was not observed, but a decrease in B. uniformis abundance was. Moreover, the abundance of B. uniformis was higher in patients with active BD than in those with diet-matched HCs. Therefore, diet change might not have been the primary factor altering the abundance of B. uniformis in the gut microbiota during BD disease activity shift. Given our previous observation that SCFA negatively affects the production of inflammatory cytokines in patients with BD (Yun et al., Ann Dermatol 2018), we speculated an alteration in the abundance of SCFA producers during BD disease activity shift.”

Reviewer 2 Report
The aim is stated clear. The authors stated clearly what study found and how they did it. The title is informative and relevant.
The references are relevant and recent. Appropriate and key studies are included. The introduction reveals what is already known about this topic.
The study methods are valid and reliable. There are enough details provided in order to replicate the study.
The data is presented in an appropriate way. The text in the results adds to the data and it is not repetitive. Statistically significant results are clear.
The conclusions answer the aim of the study. The conclusions are supported by references and own results.
Specific comments on weaknesses of the article and what could be improved:
Major points - none
Minor points
- Please, state the limitations of the study
- Could you please discuss the clinical implications of the results?
- You stated in the conclusion that may be a gut microbiota is involved in BD pathophysiology. How?
Author Response
Response to Reviewer 2 Comments
Point 1: Please, state the limitations of the study
Response 1: Thank you for your suggestion. Accordingly, we have discussed our study limitation in the revised Discussion section (page 15, lines 381-386).
“This study has some limitations. Our sample size was small owing to the very low incidence of BD. Moreover, as mentioned above, the lack of inclusion of the BD-matched HC sample at the time point of the inactive phase of BD is another limitation of this study. However, this study included dietary habit-matched family members of each patient as HC to minimize the dietary effect on the microbial composition and analysis of gut microbiome in BD patients according to disease activity.”
Point 2: Could you please discuss the clinical implications of the results?
Response 2: Thank you for your question. Although our findings suggest that the gut microbiome may be associated with BD disease activity and symptom, we did not verify the relationship between our findings and BD pathogenesis. Therefore, we have added the following to the revised manuscript (page 15, lines 390-393).
“Further animal model studies or immunologic studies are warranted to determine whether gut microbial dysbiosis may contribute to BD pathogenesis, and the modulation of the gut microbiome may be a new therapeutic target against BD.”
Point 3: You stated in the conclusion that may be a gut microbiota is involved in BD pathophysiology. How?
Response 3: As mentioned in our response to comment #2, we have revised the conclusion section (lines 390-393, page 15) and discussed the possibility of gut microbiome involvement in BD pathogenesis as follows (in the discussion section page 14, lines 352-358).
“Given our previous observation that SCFA negatively affects the production of inflammatory cytokines in patients with BD (Yun et al., Ann Dermatol. 2018), we speculated an alteration in the abundance of SCFA producers during BD disease activity shift. Our findings do not concord with this speculation but show the alteration in B. uniformis abundance, suggesting a possible role of the gut microbiome in BD pathogenesis. Further investigations regarding the role of B. uniformis in immunological and metabolic regulation are required.”

Reviewer 3 Report
Dear Authors,
Congratulations for your work which I found very interesting and well-organized. I just suggest you to improve your introduction by adding at its end the null hypotheses of the study.
Kind regards
Author Response
Response to Reviewer 3 Comments
Point 1: Congratulations for your work which I found very interesting and well-organized. I just suggest you to improve your introduction by adding at its end the null hypotheses of the study.
Response 1: Thank you for your kind comment and suggestion. Accordingly, we have added the hypotheses of our study to the revised Introduction section (lines 59-61, page 2) as follows:
“In this study, we hypothesize that the fecal and salivary microbiomes are altered in BD according to disease activity, and that this microbial dysbiosis is associated with the pathophysiology of BD.”
